# Functional connectivity in human auditory networks and the origins of variation in the transmission of musical systems

**Massimo Lumaca[1]\*, Boris Kleber[1], Elvira Brattico[1], Peter Vuust[1†], Giosue Baggio[2†]**

[1]Center for Music in the Brain, Department of Clinical Medicine, Aarhus University and The Royal Academy of Music, Aarhus, Denmark; [2]Language Acquisition and Language Processing Lab, Department of Language and Literature, Norwegian University of Science and Technology, Trondheim, Norway

**Abstract** Music producers, whether original composers or performers, vary in their ability to acquire and faithfully transmit music. This form of variation may serve as a mechanism for the emergence of new traits in musical systems. In this study, we aim to investigate whether individual differences in the social learning and transmission of music relate to intrinsic neural dynamics of auditory processing systems. We combined auditory and resting-state functional magnetic resonance imaging (fMRI) with an interactive laboratory model of cultural transmission, the signaling game, in an experiment with a large cohort of participants (N=51). We found that the degree of interhemispheric rs-FC within fronto-temporal auditory networks predicts—weeks after scanning—learning, transmission, and structural modification of an artificial tone system. Our study introduces neuroimaging in cultural transmission research and points to specific neural auditory processing mechanisms that constrain and drive variation in the cultural transmission and regularization of musical systems.

DOI: https://doi.org/10.7554/eLife.48710.001

**\*For correspondence:**
massimo.lumaca@gmail.com

[†]These authors contributed equally to this work

**Competing interests:** The authors declare that no competing interests exist.

## Introduction

Human brain function shows substantial variability across individuals (*Bergfeldt, 2016*). An emerging question in the study of cultural evolution is whether aspects of variability in brain function relate to variation in the acquisition and transmission of symbolic systems observed across individuals in a population (*Dediu and Ladd, 2007*). In recent years, it has been argued that symbolic systems, including language and music, evolve while they are being transmitted across individuals and generations, partly adapting to properties of brain systems of learners and users (*Deacon, 1997*; *Christiansen and Chater, 2008*). This view has found support in laboratory experiments on the cultural transmission of language and music (*Kirby et al., 2008*; *Ravignani et al., 2016*; *Lumaca and Baggio, 2017*). These studies showed that novel and initially unstructured (artificial) signaling systems become gradually more organized during transmission along chains of individuals ('generations'). Artificial symbolic systems are generally assumed to be relevant to understanding the cultural transmission and evolution of natural symbolic systems (*Scott-Phillips and Kirby, 2010*). One major unresolved question is whether variability in brain structure and function can be linked to interindividual differences in learning, retaining, and transmitting symbolic material. To our knowledge, no study has addressed this question using neuroimaging techniques.

We investigate this issue in the music domain by examining a core functional property of brain systems: resting-state functional connectivity (rs-FC). Rs-FC is a measure of the temporal correlation of spontaneous slow fluctuations (<0.1 Hz) in the BOLD signal between spatially separated brain regions, derived from resting-state fMRI (rs-fMRI) data (*Biswal, 2012*). This measure is receiving

much attention because it captures spontaneous interactions between neuronal populations in functionally associated networks (*Damoiseaux et al., 2006*), and it has been recognized as a promising tool for assessing individual differences in brain function (*Dubois and Adolphs, 2016*).

One hypothesis in the field of bio-musicology is that some musical traits and behaviors may be traced back to the local functioning of the auditory cortex (*Trainor, 2015*; *Lumaca et al., 2018b*). With each generational transfer, music material has to pass through an information processing 'bottleneck' that constrains music perception and acquisition (*Merker et al., 2015*). Recent evidence indicates that this bottleneck can be instantiated in anatomical and functional connectivity patterns of the auditory system and other (e.g. attentional and motor) brain systems (*Zatorre, 2013*). For example, a stronger functional connectivity between left and right auditory cortex, and between auditory cortex and prefrontal areas, as measured during active listening, tends to be associated with facilitation in the encoding and discrimination of melodic material in experimental participants (*Andoh and Zatorre, 2011*; *Andoh and Zatorre, 2013*). When measured at rest, functional connectivity networks may serve as indicators of auditory learning skills (*Ventura-Campos et al., 2013*; *Veroude et al., 2010*) and have been shown to be plastic and music-experience dependent (*Fauvel et al., 2014*; *Tanaka and Kirino, 2016*; *Zamorano et al., 2017*). Here, we address the specific question whether spontaneous functional connectivity patterns within portions of the auditory cortex, and between the auditory cortex and other relevant (e.g. prefrontal and parietal) networks, as measured by rs-fMRI, can drive and predict inter-subject variability in three core aspects of cultural transmission: (1) social learning, (2) transmission, and (3) structural modification of an unfamiliar (artificial) tone system. We use the term 'prediction' to indicate that a brain measure collected at one point in time in one group may be used to predict a behavioral outcome at a later point in time in the same group, without however implying out-of-sample generalization (*Gabrieli et al., 2015*).

In our study, we modeled these three behaviors using a laboratory model of interactive cultural learning and transmission, the signaling game (*Moreno and Baggio, 2015*; *Nowak and Baggio, 2016*; *Baggio, 2018*). In signaling games (*Lewis, 1969*; *Skyrms, 2010*), a sender and a receiver exchange signals over subsequent rounds with the aim of developing a 'common code': here, a finite set of tone sequences associated with specific semantic or affective states. In previous work (*Lumaca and Baggio, 2016*), we showed that individual players' ability to learn and transmit an artificial tone system in signaling games, and the extent to which structural aspects of the codes are reorganized by players, can be predicted based on the latency of a neurophysiological marker of auditory processing in participants: the mismatch negativity (MMN) (*Näätänen et al., 1978*). Complex cultural phenomena, such as the transmission of musical systems (*Le Bomin et al., 2016*), may therefore be partly constrained by the functional organization of the auditory cortex in the human brain.

We assessed this hypothesis by combining resting-state and auditory fMRI and signaling games in two sessions (20 days apart on average). In the first session, each participant was first scanned at rest (rs-fMRI) and then subjected to functional scanning to localize auditory networks using an auditory oddball paradigm (*Lumaca and Baggio, 2016*) (*Figure 1*). The use of a functional localizer independent of the behavioral task has at least one major advantage: it allows us to test specific hypotheses of brain function in a statistically unbiased dataset, increasing the generalizability of findings (*Saxe et al., 2006*). In the second session, each participant played two signaling games, the first as receiver (or learner), and the second as sender (transmitter). In Game 1, participants had to learn from a confederate of the experimenters a melodic signaling system of the same kind used by *Lumaca and Baggio (2016)*. In Game 2, they had to transmit the code they had learned in the previous game to a receiver. We tested the hypothesis that rs-FC between auditory regions identified with a functional localizer task in session one predicts—weeks after scanning—learning, transmission, and structural modification (*Figure 2*) of the signaling system used in the games of session 2.

## Materials and methods

### Participants

Fifty-two right-handed volunteers (33 females, mean age 24.5 years, range 20–34) with normal hearing participated in the study (two additional participants did not go through the entire experimental MRI session since they experienced discomfort). Fifty-one of them completed the entire experiment:

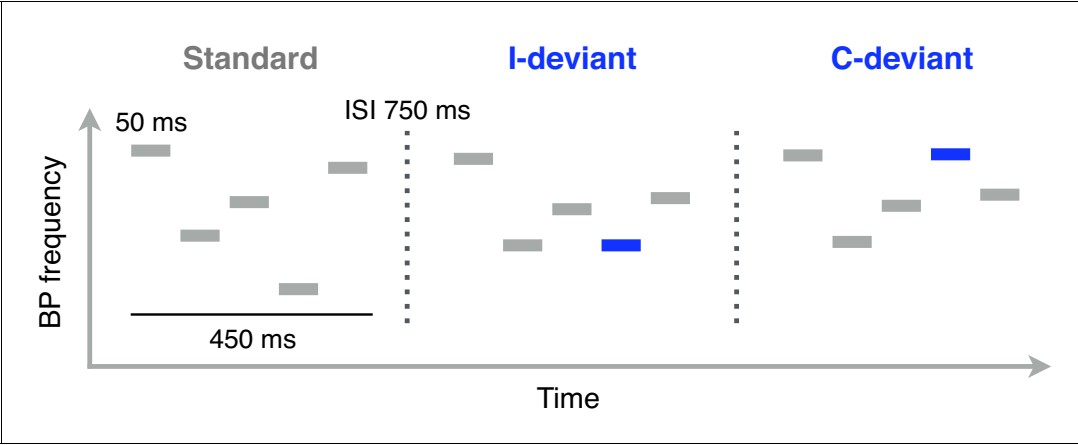

**Figure 1.** Schematic illustration of Bohlen-Pierce (BP) tone sequences used in the functional localizer task (auditory oddball). Functional MRI scanning was performed while participants listened to these sequences. Each sequence consisted of 5 sinusoidal 50 ms tones separated by 50 ms of silence. The intersequence interval (ISI; the silent gap between the offset of one sequence and the onset of the next one) was 750 ms. Standard sequences (80% of trials) were randomly transposed at different registers. Deviant sequences featured a change in pitch interval (interval deviants, I-deviants; 10%) or melodic contour (contour deviants, C-deviants; 10%) of the fourth tone, relative to its position in standard sequences.
DOI: https://doi.org/10.7554/eLife.48710.002

one person (female) dropped out before the behavioral part, due to reasons unrelated to the study. Participants were all musically naive individuals (i.e., none of them had three or more years of formal musical training; mean 0.6 years ± 0.98). All participants gave their written consent and filled out an fMRI safety form before the experiment. The study was approved by the local ethics committee of the Central Denmark Region (nr. 1083).

## Study design

Each participant took part in two sessions occurring on two separate days (mean 20 days apart, range 13–30 days). The MRI session on day 1 consisted of 3 main scans, including: one resting state run, where the subject was instructed to keep her eyes closed and let her thoughts wander; one run of acquisition of high-resolution anatomical images (T1-weighted); and one functional localizer run, consisting of a passive auditory oddball task (*Figure 1*), to identify brain regions preferentially responding to automatic detection of deviant sounds (see 'Session 1: MRI'). At the end of each MRI session, the short-term and working-memory spans of participants were measured using a forward and backward digit-span test (*Orsini et al., 1987*).

The second session consisted of a cultural transmission experiment (*Figure 3*), in which each participant played two fixed-roles signaling games sequentially, with a confederate of the experimenter: in the first game as receiver (Game 1) and in the second as sender (Game 2) ('Session 2: Signaling Games') with fixed sender and receiver roles throughout each game. In Game 1, the participant learned from the confederate (sender) a system of tone sequences associated to emotions. In Game 2, the participant (now the sender) was instructed to transmit the melodic system they had learned to the receiver. Three confederates of the experimenters were randomly assigned to the games and were all blind with respect to the starting material.

## Bohlen-Pierce scale

The tone sequences used in the oddball task and in the behavioral experiment were built using pure tones from the Bohlen-Pierce (BP) scale (*Mathews et al., 1988*). In the equal-tempered version of the BP scale, a tritave (3:1 frequency ratio) is divided into 13 logarithmically even steps, and the frequency (F) of each tone is given by the formula $F = k \times 3^{(n/13)}$, where $n$ is the number of steps, and $k$ is the fundamental frequency (see below).

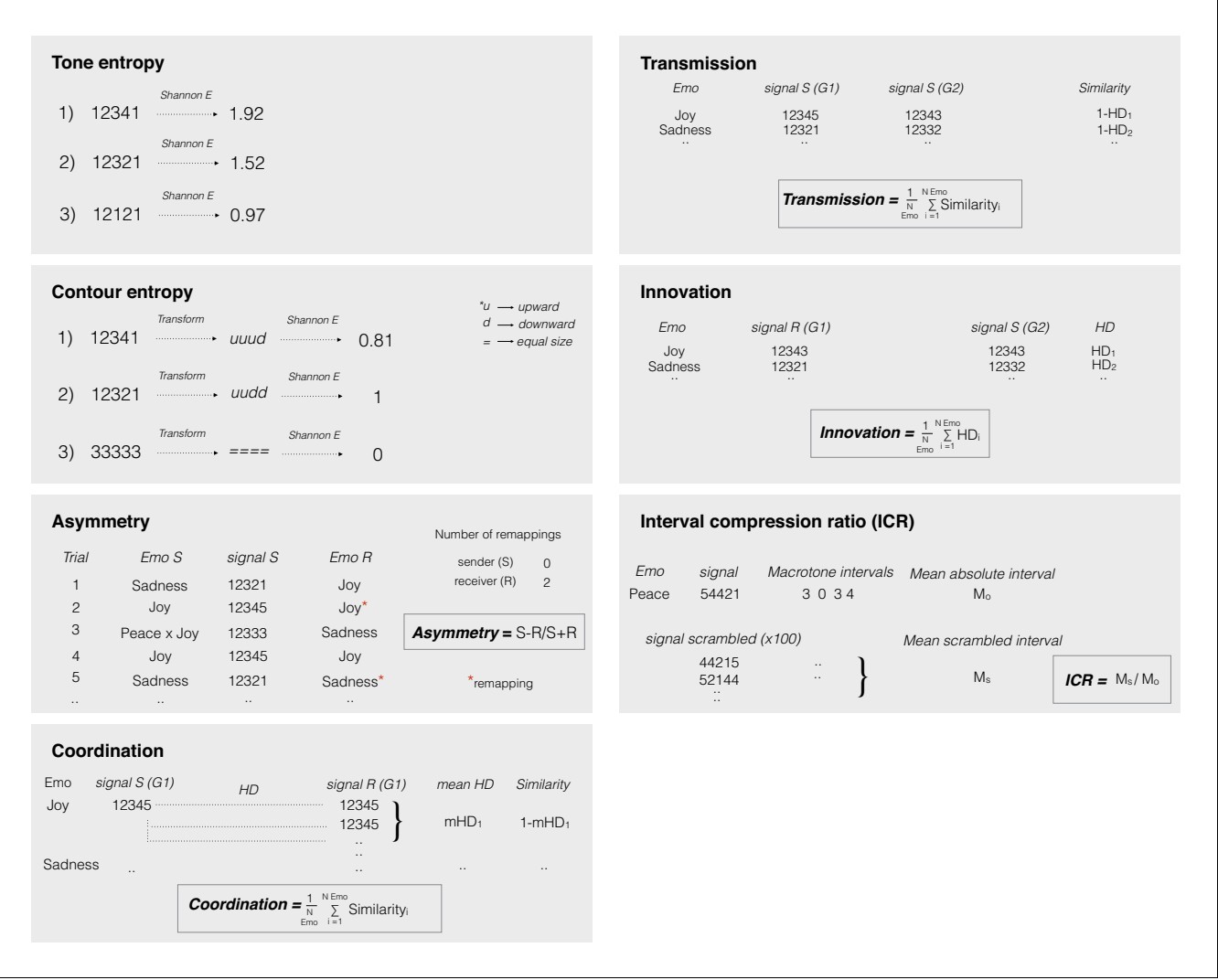

**Figure 2.** Schematic example of the formal measures used on signaling game data. *Tone and Contour entropy:* A single entropy value (**H**) was computed for each tone sequence (Tone entropy) and the relative contour transform (Contour entropy) by using the Shannon entropy formula. These measures are different indicators of the complexity of a melodic signal. *Asymmetry:* we calculated asymmetry in Game 2 as the difference of the number of code changes introduced by the sender (**S**) and by the receiver (**R**) across all emotions, divided by the total number of code changes ((S–R)/(S+R)). This measure reflects the direction of information flow and ranges from −1 (the receiver adapts his mappings to the ones used by the sender) to 1 (viceversa). In Game 1, the sender never changes the signal-to-emotion mappings and asymmetry is −1 by design. *Coordination:* we computed coordination in Game 1 as the mean similarity (1-Hamming distance) between the signal (tone sequence or contour transform) used by the sender for a given emotion and the set of signals mapped by the receiver to the same emotion during the second half of the game (between-player measure). A mean value was computed across all five emotions. It represents the extent to which the sender and receiver shared their mappings at the end of the game and it ranges from 0 (different signaling system) to 1 (shared signaling system). *Transmission:* this measure was computed as the similarity of signals (tone sequences or contour transforms) mapped to the same emotion and used by senders of adjacent games (between-player measure). A mean value was computed across emotions. It ranges from 0 (the sender in game two reproduced a new signaling system) to 1 (the sender in game two faithfully transmitted the signaling system received in game 1). *Innovation:* this measure was calculated as the mean hamming distance (HD) between the signal (tone frequency or contour transform) mapped with the greatest frequency to a given emotion in the second half of Game 1 by the player as receiver and the signal reproduced with major frequency for the same emotion in Game 2 by the same player as sender (within-player measure). A mean value was computed across the five emotions, ranging from 0 to 1. Values close to one indicate that a significant number of changes were introduced in the signaling system by the participant. *Interval compression ratio:* The interval compression ratio (ICR) (*Tierney et al., 2011*) was calculated as the ratio between the mean absolute interval size (in macrotones) of a tone sequence randomly scrambled ($N = 100$), divided by the mean absolute interval size of the original sequence. A mean value was computed across the five emotions. Larger values indicate a bias towards smaller (proximal) intervals. Scripts to compute these measures are available at: https://doi.org/10.5061/dryad.2jj01c1 (folder 'signaling games').

DOI: https://doi.org/10.7554/eLife.48710.004

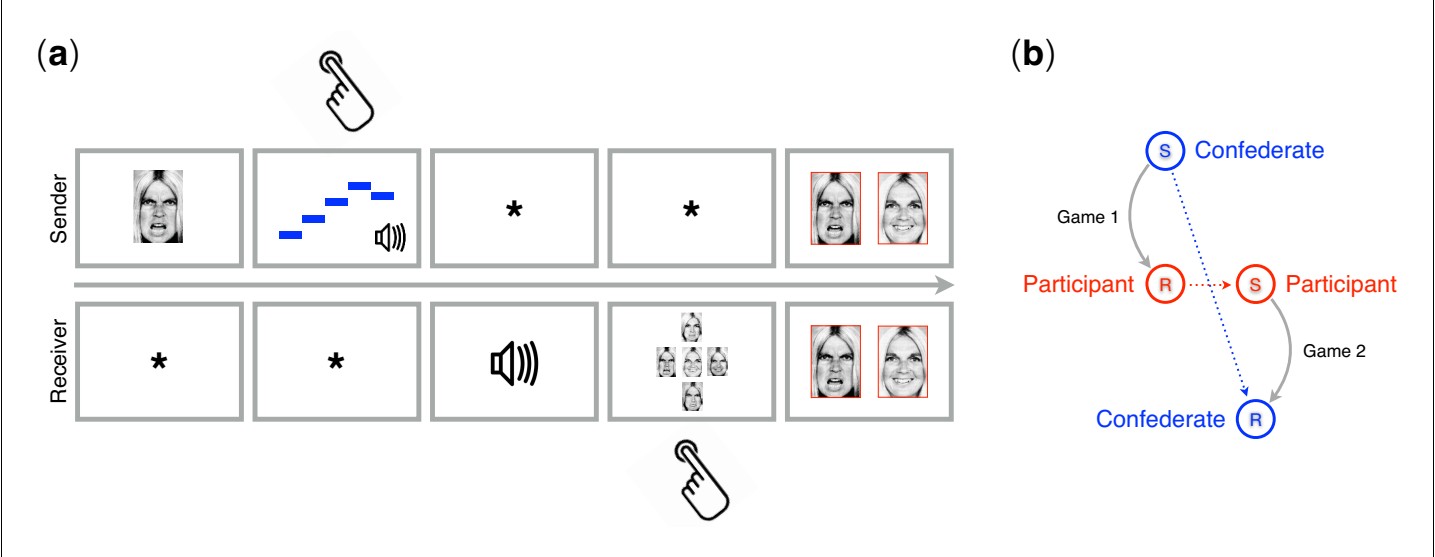

**Figure 3.** Trial structure and experimental transmission design. (a) Example of a trial from the signaling games played by participants in the second session of the study. The top and bottom rows show what senders and receivers saw on the screen, respectively. For the sender, the task was to compose an isochronous five-tone sequence to be used as a signal for the simple and compound emotions expressed by the image shown on the screen at the start of each trial. For the receiver, the task was to respond to that signal by guessing the image the sender had seen. The sender and the receiver converged over several trials on a shared mapping of signals (tone sequences) to meanings (emotions). Hand symbols indicate when the sender or the receiver had to produce a response. Feedback was provided to both players simultaneously, showing the face seen by the sender and the face selected by the receiver in a green frame (matching faces; correct) or in a red frame (mismatching faces; incorrect). Time flows from left to right. (b) Experimental transmission design in signaling games. The participant played as receiver (R) with a confederate of the experimenters playing as sender (S) in Game 1. Roles switched in Game 2.

DOI: https://doi.org/10.7554/eLife.48710.003

### Session 1: MRI
#### Oddball stimuli and task

We adopted the same stimuli and task as used in *Lumaca and Baggio (2016)* (*Figure 1*). Sequences consisted of five 50 ms sinusoidal tones (5 ms rise and fall) from the frequencies 440, 521, 567, 617, 730.6 Hz (low register). Each tone was followed by 50 ms of silence with an ISI (intersequence interval) of 750 ms. To avoid adaptation effects, the stimuli were randomly transposed at different registers of the BP scale (lowest frequencies: 440, 478, 567 Hz).

In a single run, participants were presented with repeated Standard sequences (80% of the trials), interspersed with Contour (10%) and Interval deviant sequences, adding up to 1260 stimuli. In contour deviants, the fourth tone violated the surface structure ('ups' and 'downs') of a sequence, but not the interval size; vice versa for the interval deviants. To obtain a stronger oddball BOLD response, two deviant sequences of the same type (contour or interval) were always presented in close temporal succession (range 2400–4800 ms).

#### MR image acquisition

Whole-brain data were acquired on a 3T MRI scanner (Siemens Prisma). Participants laid in supine position in the scanner, with cushions fit around their heads to reduce motion artifacts. Participants were instructed to keep their eyes closed during the rs-fMRI (resting state) scan, and they watched a subtitled silent movie during the subsequent oddball scan. During the fMRI oddball scan, the participant was asked to ignore the auditory stimulation and focus on the movie. The movie was shown on an MRI-compatible screen, located at the rear of the scanner, viewed by participants through a mirror mounted on the head coil. All stimuli were presented by MRI-compatible headphones using Presentation software (https://www.neurobs.com/).

For the rs-fMRI (resting state), a total of 600 volumes were acquired over 10 min using fast T2*-weighted echo-planar imaging (EPI) multiband sequence (TR, 1000 ms; TE, 29.6 ms; voxel size, 2.5

mm$^3$). For the functional localizer task, a total of 1535 volumes were acquired over 25 min using the same multiband EPI sequence (TR, 1000 ms; TE, 29.6 ms; voxel size, 2.5 mm$^3$). High-resolution T1-weighted images (TR, 5000 ms; TE = 2.87 ms; voxel size, 0.9 mm$^3$) were obtained using an MP2RAGE sequence.

## Preprocessing and analysis of auditory functional localizer data

Pre-processing and statistical analyses of fMRI data were performed using SPM12 (r7487), implemented in Matlab R2016b (Mathworks), and included all 52 participants. During preprocessing, functional data (EPI) of each subject were realigned to the first image of the session to correct for head motion. High-resolution anatomical images of individual subjects were coregistered to the mean functional EPI and segmented using the default SPM12 settings. The resulting deformation fields were used to normalize the functional images to standard Montreal Neurological Institute (MNI) space, resampled to 2 mm$^3$ voxels, and spatially smoothed by means of an isotropic Gaussian kernel of 6 mm full-width at half-maximum (FWHM).

To remove low-frequency noise, the data were high-pass filtered (cutoff 1/128 Hz) and corrected for serial autocorrelation using an AR(1) function. At the single-subject level, we set up a general linear model (GLM) including three experimental regressors, standard (STD), deviant contour (C-deviant), and deviant interval (I-deviant), convolved with a canonical hemodynamic response function (HRF), plus the six rigid-body parameters (three rotational, three translational) to account for head motion. To obtain a balanced contrast between standards and deviants in the second-level analysis, we randomly extracted a number of standard events (onsets of the fourth tone) equal to the number of deviant events (n = 126), with the only constraint that standard patterns had to be preceded and followed by at least five other standard patterns. These standard events were implicitly modeled in the GLM design matrix. The remaining standard events (STD) (n = 882) were explicitly modeled in the GLM matrix, together with deviant contour events (C-deviant) (n = 126) and deviant interval events (I-deviant) (n = 126). At the group level, we used a whole-brain random effects analysis using *t*-test contrasts for C-deviant >STD and I-deviant >STD. Whole-brain significant activations were identified at p<0.001, familywise error (FWE) corrected at voxel-level.

## Seed region definition

Different regions of interest (ROIs), or seeds, were constructed as 5 mm spheres centered on the peak voxels from the auditory functional localizer task. Following this procedure, four ROIs were generated, centered on the following MNI coordinates (*x y z*): [54 2 -4], [−52–14 4], [66 -16 4], [−66–22 6]. These seeds were used in rs-FC (resting-state functional connectivity) and psychophysiological interaction (PPI) analyses (details below), and to extract task-based activation changes from functional localizer data. To this end, mean percent signal changes were extracted from each ROI and condition (STD, C-deviant, I-deviant) using MarsBaR Toolbox (http://marsbar.sourceforge.net/; *Brett et al., 2002*).

## Preprocessing and analysis of rs-FC data

Resting-state functional connectivity (rs-FC) preprocessing and analysis were implemented by using a standard pipeline in the SPM CONN toolbox (*Whitfield-Gabrieli and Nieto-Castanon, 2012*). Preprocessing included: realignment, direct segmentation and normalization to the MNI space (2 mm$^3$), outlier detection (ART-based identification of outlier scans for scrubbing; motion correction = 0.9 mm; global-signal *z*-value threshold = 5) (https://www.nitrc.org/projects/artifact_detect), and smoothing (FWHM = 6 mm). The realignment and scrubbing parameters and the BOLD signal from white matter and cerebrospinal fluid were regressed out in the GLM. Data were band-pass filtered at 0.008–0.09 Hz to reduce the effects of low-frequency drifts and high-frequency noise.

## Neural predictors of behavioral performance

Using a seed-based correlation analysis (SCA) approach, for each participant, we first tested the functional relationship between the seeds identified in the localizer task. The mean time course of all voxels within each seed was used to calculate individual ROI-to-ROI pairwise linear Pearson's correlations, and the resulting *r* values were normalized to *z* values via Fisher's *z*-transformation. Pairwise correlations were computed for all ROIs. From each participant we obtained 6 *z* values, each

reflecting the degree of connectivity between any pair of ROIs. We computed Pearson's correlations between Fisher's *z* values and performance measures in signaling games (see below). Second, using CONN, we performed whole-brain FC analyses separately for each auditory seed. In this analysis, we explored whether the rs-FC between auditory ROIs and other (e.g., parietal and prefrontal) brain areas could predict behavioral performance. Single-subject seed-to-voxel correlation maps were calculated by extracting the residual BOLD time course from each seed, and by computing Pearson's correlation coefficients between that time course and the time course of all other voxels in the brain. Correlation coefficients were normalized to *z* values via Fisher's *z*-transformation. At group-level, we implemented a linear regression analysis for each seed, by defining the simple main effect of performance as between-subject contrast. Statistical thresholds were set to $p < 0.001$ (uncorrected) at single voxel level, and the resulting cluster(s) were thresholded to a cluster-mass FWE $p < 0.05$ using non-parametric permutation testing two-tailed ($n = 1000$).

To examine whether task-dependent functional connectivity was related to signaling game performance, we also conducted a PPI analysis (*Friston et al., 1997*). We tested whether oddball-evoked changes in the functional connectivity of the same auditory network could predict behavioral performance and structural modification in signaling games. Volumes of interest (VOIs) were created as 5 mm spheres centered on single-subject global maxima for the contrast of interest (C-deviant >STD: $p < 0.05$ uncorrected; minimum voxel extent 3). This subject-specific global maximum was identified within a sphere of 10 mm radius centered on the peak of the group effect (C-deviant >STD). In participants where no active cluster could be found, the statistical threshold was lowered to $p < 0.5$ uncorrected. The BOLD signal was summarized in each participant using the first eigenvariate (principal component) of suprathreshold voxels within the VOIs. The PPI design matrix contained the following three variables: (1) a 'psychological' variable that represents the contrast of interest (C-deviant >STD), (2) a 'physiological' variable that represents the time-course of the seed region, and (3) an interaction term between the psychological and physiological variables. Individual β images of the interaction term were entered into a whole-brain second-level regression analysis (*Ofen et al., 2012*) with each behavioral variable as a covariate of interest—that is one behavioral variable at a time. We constrained the whole-brain regression analysis to a 'target' region—a spherical binary mask of 5 mm radius centered on the peak of the group effect. In line with previously described resting-state analyses, we performed a regression analysis for each possible seed-to-target combination.

## Session 2: Signaling games

### Signaling games

In the version adopted here, signaling games are cooperative games of incomplete information (*Lewis, 1969*), where a sender and a receiver exchange signals without a prior agreement on pairings of signals to meanings: players should then agree, over multiple signaling rounds, on a 'common code' (i.e. a shared set of signals-meaning mappings). A typical interaction may consist in the sender privately witnessing a state of affairs, which must be communicated to the receiver using a signal. Upon receiving the signal, the receiver must take appropriate action to match the initial state of affairs. Feedback may follow, providing players with information about whether they associated the signal with corresponding states and actions.

### Procedure and stimuli

shows an example of a single signaling round or trial. A facial expression (state) is privately displayed on the screen to the sender, who is in turn instructed to produce a five-tone pattern (signal) matching the observed event. The signal is produced using digits (1-5) of the computer keyboard and is sent to the receiver. The receiver listens to the signal through headphones and is asked to select one of the five facial expressions (i.e., the one he believes the sender has seen) by using the keyboard's numeric keypad. A feedback (3 s) is then provided to both participants, showing on their screen the state as seen by the sender and the one chosen by the receiver. There was no reward for speed, either in a trial or in the game. Successful coordination was the only reward for participants. In this study, player roles were fixed throughout a game, which consisted of several trials (see below). *Moreno and Baggio (2015)* demonstrate that, with fixed roles, it is (mostly) the receiver who modifies her code to adapt to the sender. Asymmetry in the information flow from senders to receivers is a core property of cultural transmission. The signals were five isochronous melodic

patterns of 5 sinusoidal tones, with each tone drawn from the BP scale ($k$ = 440 Hz; $n$ = 0, 4, 6, 7, 10); the states were five emotions, simple (peace, joy, sadness) and compound (peace X joy; peace X sadness), shown as facial expressions of one actor (for more details, see *Lumaca and Baggio, 2016*; *Lumaca and Baggio, 2017*).

## Transmission design

Each participant played sequentially two signaling games with a confederate of the experimenters, changing his/her role from the first game (receiver) to the next (sender) (*Figure 3b*). Players converged on (i.e. learned or transmitted) a melodic signaling system, that is a set of melodic patterns with mappings to emotions. At the start of Game 1, one of two sets of stimuli (starting material) was delivered by the confederate player. The confederate was instructed to press, on the computer keyboard, a numeric digit showed on the screen (1 to 5). Each key was mapped to one of the five melodic patterns of the seeding material: in Game 1, he played with no sound feedback (volume set at 0), to prevent him from learning the starting material and be biased in Game 2. The confederate was instructed to use consistently that mapping throughout Game 1, regardless of what the participant (receiver) would do. In each set, signals had different melodic contour complexity: two high-entropy (Shannon Entropy, H = 1 bit), two low-entropy (H = 0.81 bits), and a monotone stimulus (control; H = 0 bits) (details below). The participant was instructed to learn the signaling system in Game 1 and to re-transmit it as faithfully as possible to the receiver in Game 2 (see Procedure and stimuli). The starting material was counterbalanced across participants. Each participant played 70 trials in Game 1 and 30 trials in Game 2.

## Data analysis

The aim of this study is to investigate whether individual differences in the intrinsic connectivity between regions of auditory networks, identified by means of an auditory-cortex functional localizer for melodic processing, can predict various aspects of the evolution of an auditory signaling system: coordination, learning, transmission, code innovation, and structural modification (*Figure 2*) (see below for details). A Pearson's correlation analysis was conducted between the ROI-to-ROI $z$ values obtained by rs-FC (i.e., from each of the six possible ROI-to-ROI combinations), the digit span scores, and information theoretic measures of cultural behavior (i.e., edit distance and entropy), as observed in signaling games. All p-values were Bonferroni corrected at $\alpha$ = 0.05/7 = 0.007 (7 is the number of independent tests involving each behavioral variable) (*Table 1*).

## Signaling measures

We used Hamming distances (*Hamming, 1986*) as a measure of degrees of cross-player coordination, code transmission, and innovation in signaling games. Hamming distance is the number of pointwise substitutions (S) necessary to make two strings of equal length (L) identical, and the normalized value [0 to 1] is given by S/L. Values close to one are associated to different strings. The strings compared were pairs of patterns of five tones (tone distance) or of four interval directions (i.e. ups and downs) between adjacent tones (contour distance). Coordination (as a learning index) was measured as tone and contour similarity (i.e. inverse Hamming distance) between the signals used by the confederate (sender) and the signals most frequently paired with corresponding emotions during the second half of Game 1 by the participant (receiver). A value close to one indicates that the melodic material was effectively learned by the participant during Game 1. Transmission was calculated as the similarity between the signals used by senders of adjacent generations (between-player measure). A value close to one indicates an accurate recall of the initial melodic material by the participant during Game 2. Innovation was measured as the distance between signals learned by the participant in one game as receivers and transmitted in the next game as senders (within-player measure). Furthermore, an asymmetry index A=(S-R)/(S+R), measuring the direction of information flow, was computed. In the equation, asymmetry (A) is the difference between the code changes made by the sender (S) and those made by the receiver (R), divided the total number of changes. Finally, we used the number of correct trials in game one as a measure of individual performance accuracy.

**Table 1.** Pearson product-moment correlations (*r*) between neural predictors (Fisher's z-transformed ROI-to-ROI rs-FC values), neuropsychological predictors (digit span), and behavioral measures related to learning (coordination, transmission, innovation, accuracy) and structural regularization (proximity, melodic regularization).

| | | lHG-rHG | lHG-lSTG | lHG-rSTG | rHG-lSTG | rHG-rSTG | lSTG-rSTG | Digit Span |
|---|---|---|---|---|---|---|---|---|
| *Learning* | | | | | | | | |
| Coordination | Tone | 0.05 | 0.26 | 0.10 | 0.17 | 0.12 | **0.41** | 0.22 |
| | Contour | 0.15 | 0.33 | 0.13 | 0.10 | 0.02 | 0.35 | 0.11 |
| Transmission | Tone | -0.09 | 0.23 | 0.01 | 0.13 | 0.14 | **0.41** | 0.11 |
| | Contour | -0.08 | -0.04 | -0.05 | -0.10 | -0.04 | 0.009 | 0.008 |
| Innovation | Tone | -0.01 | **-0.41** | -0.10 | -0.20 | -0.12 | **-0.40** | 0.05 |
| | Contour | -0.03 | -0.10 | -0.05 | 0.01 | 0.004 | -0.04 | 0.08 |
| Accuracy | | -0.08 | 0.21 | 0.04 | 0.12 | -0.01 | **0.36** | 0.17 |
| *Structural regularization* | | | | | | | | |
| Proximity | | 0.33 | 0.07 | 0.01 | 0.21 | 0.04 | -0.05 | -0.17 |
| Melodic Regularization | Tone | 0.20 | -0.07 | -0.05 | 0.03 | -0.09 | -0.12 | -0.14 |
| | Contour | **0.43** | 0.08 | 0.21 | 0.29 | 0.12 | 0.02 | -0.16 |

*Notes:* Correlation coefficients marked in bold are significant under Bonferroni correction ($\alpha$=0.05/7 = 0.007; seven is the number of independent tests on each behavioral variable). r = right hemisphere; l = left hemisphere. STG = superior temporal gyrus; HG = Heschl's gyrus; Tone = measure computed using tone distance; Contour = measure computed using contour distance.

DOI: https://doi.org/10.7554/eLife.48710.005

## Entropy

The complexity of melodic signals (and of their contour) was measured using Shannon Entropy (**Shannon, 1948**):

$$H(X) = -\Sigma \, P(x_i) \log_2 p(x_i)$$

where *X* is the pattern of five tones (sequence complexity) or the four interval directions (contour complexity), and $p(x_i)$ the probability of the *i*th outcome. A mean entropy value was calculated separately for the tone sequences and their contour transforms, for Game 1 and Game 2. The difference in this value between Games 1–2 was used as a measure of regularization. A positive value indicates regularization made by the participant on the initial melodic material (or its contour transforms) during recall in Game 2.

## Pitch proximity

A key property of melodic structure is pitch proximity. It represents the tendency of small pitch intervals to outnumber large ones in music (**Savage et al., 2015**). The interval compression ratio (ICR) was used here as a measure of melodic proximity (**Tierney et al., 2011**), and it is computed as follows: first, all the tones of a sequence are randomly shuffled, and a mean absolute interval is computed for the resulting sequence; this procedure is repeated 100 times; a mean of these averages is computed and is used as the numerator of the ICR equation. The denominator is the mean absolute interval of the original sequence. Large values indicate a bias towards small intervals. The difference in this value between Games 1–2 was used as a measure of regularization toward smaller melodic intervals.

# Results

## Signaling games

Our results indicate a net flow of information from senders to receivers across games, with more efficient transmission of contour than interval information. Asymmetry in Game 1 was −1 by design: following the experimenter's instructions, the confederates never adjusted their code to the one used by the participants (receivers). Asymmetry in Game two was negative and significantly different from zero (one sample Wilcoxon signed rank test: median −0.4, n = 51, Z = −6.05, p<0.001): there was a tendency for participants (now senders) to maintain their code and for receivers to adjust more frequently their mappings during coordination than senders did. Coordination in Game one was positive and significantly different from 0 using both tone distance (median 0.76) and contour distance (one sample Wilcoxon signed rank test: median 0.84, n = 51, Z = 6.21, p<0.001). Transmission was positive and different from 0, with better accuracy observed using contour distance (one sample Wilcoxon signed rank test: median 0.65, n = 51, Z = 6.22, p<0.001) than tone distance (one sample Wilcoxon signed rank test: median 0.36, n = 51, Z = 6.21, p<0.001). The initial material was successfully learned and then transmitted by all participants. Innovation was positive and different from 0, and it was higher for tone distance (one sample Wilcoxon signed rank test: median 0.64, n = 51, Z = 6.22, p<0.001) than for contour distance (one sample Wilcoxon signed rank test: median 0.64, n = 51, Z = 6.16, p<0.001).

Melodic material was reorganized from Game 1 to 2. In Game 2, melodic signals were made less entropic, especially in their melodic contour (Wilcoxon signed rank test: code regularization, median mean change 0.08, n = 51, Z = 1.60, p=0.1; contour regularization, mean change 0.07, n = 51, Z = 4.42, p<0.001) and were biased toward smaller intervals (mean ICR change 0.15, n = 51, Z = −3.92, p<0.001).

## Auditory-cortex functional localizer

Whole-brain analysis conducted for the contrast between deviant contour and standard stimuli (DC >SS) revealed sharp activations ($p_{FWE}$ <0.001) in regions of the left and right superior temporal gyrus (Table 2). Specifically, the analysis revealed four distinct clusters of activation in the bilateral STG: two anterior clusters, localized in Heschl's gyrus (HG; cytoarchitectonic areas Te1 and Te1.2, in the left and right hemisphere, respectively), and two more posterior, localized in the planum temporale (PT; cytoarchitectonic area Te3) (Morosan et al., 2005) (Figure 4). Conversely, no significant activations were detected for the contrast between interval deviant and standard stimuli (I-deviant >SS). Accordingly, only peak activations DC >STD were used to define the ROIs for the correlation analysis between rs-FC and behavioral data.

## ROI-to-ROI resting state FC results

Figure 5 shows the correlations between rs-FC and behavioral measures (coordination, transmission, innovation, accuracy). Individuals with stronger inter-hemispheric rs-FC between ROIs of lateral STG displayed better learning performance and less innovation. Fisher's z values were positively correlated with coordination (r = 0.41; p=0.002), accuracy (r = 0.36; p=0.007), and transmission (r = 0.41;

**Table 2.** Brain regions activated in the C-deviant >STD contrast (Height threshold: T = 6.76, $p_{FWE}$ <0.001; Extent threshold: k = 0 voxels).

| T statistic | MNI peak activation coordinates | Number of active voxels in the cluster | Anatomical region | Probabilistic atlas[1] |
|---|---|---|---|---|
| 8.83 | [54 2 -4] | 54 | rSTG | Area TE (1.2) 28% OP4 (PV) 17% |
| 8.54 | [66 -16 4] | 60 | rSTG | Area TE (3) 57% |
| 7.93 | [−52–14 4] | 26 | lSTG | Area TE (1) 46% Area TE (1.2) 13% |
| 7.88 | [−66–22 6] | 32 | lSTG | Area TE (3) 73% |

*Notes:* Negative coordinates denote left-hemispheric regions. C-deviant = contour deviant; STD = standard; r = right hemisphere; l = left hemisphere. STG = superior temporal gyrus. FWE = familywise error corrected. [1]Anatomical classification using the SPM anatomy toolbox (*Eickhoff et al., 2005*).
DOI: https://doi.org/10.7554/eLife.48710.007

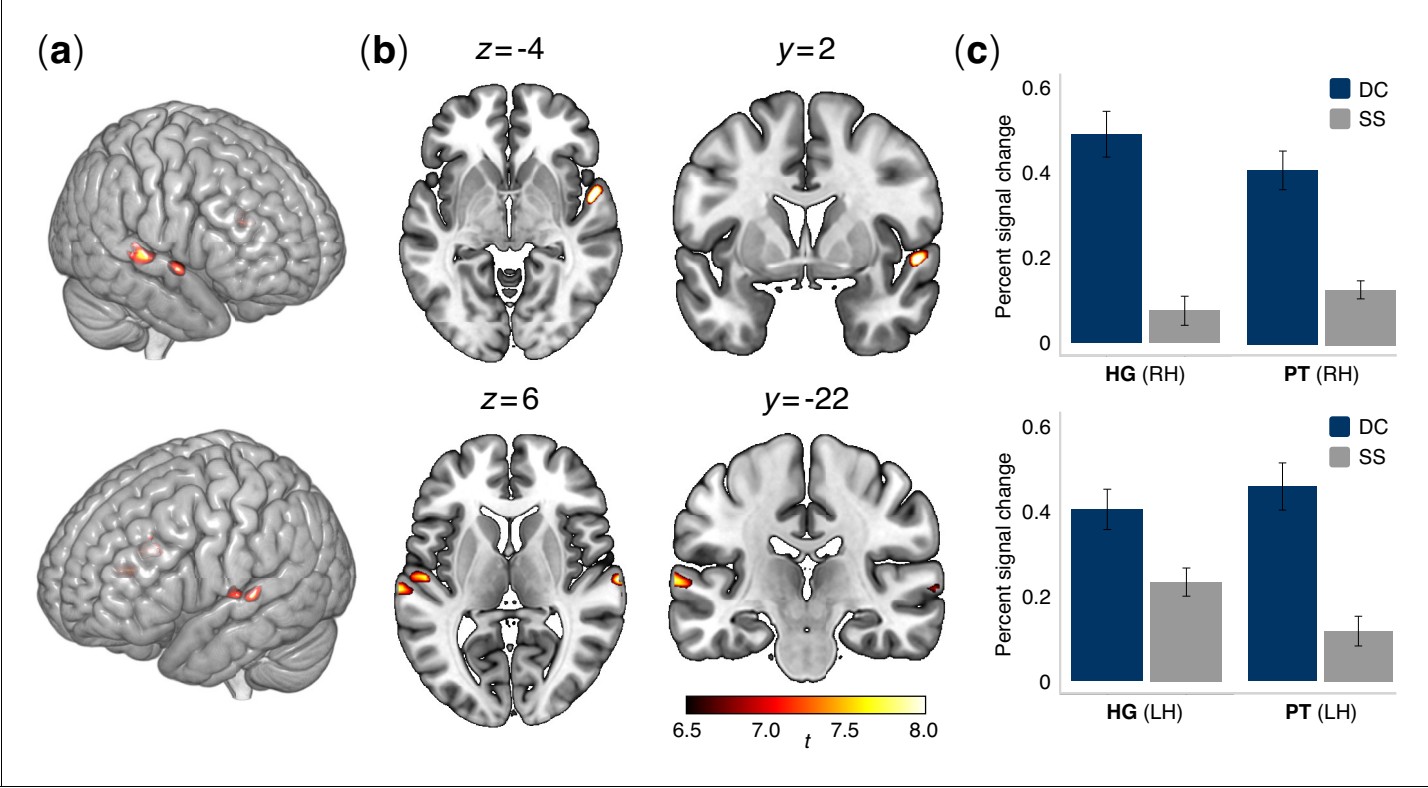

**Figure 4.** Brain activation patterns in the auditory functional localizer task. (a) Lateral and (b) axial and coronal views of active voxels in temporal cortex overlaid onto an MNI standard brain for the contrast between contour deviant and standard stimuli (DC > SS). Significant activations were observed in the bilateral posterior superior temporal cortex, specifically in Heschl's gyrus (HG) and the planum temporale (PT) bilaterally (see *Table 2*) (N = 52 participants). Contrasts were family-wise error (FWE) corrected for multiple comparisons at α = 0.001. Colormap intensities indicate indicate *t*-values. (c) Bar plots show percent changes in BOLD signal in HG and PT of the right hemisphere (upper panel) and left hemisphere (lower panel) in the comparison between contour deviants (DC) and standard sequences (SS). Error bars indicate the standard error of the mean.

DOI: https://doi.org/10.7554/eLife.48710.006

p=0.002) and negatively correlated with innovation (r = -0.41; p=0.003). A negative correlation was also observed between innovation and rs-FC between left STG and left HG (r = −0.41; p=0.002).

A similar pattern was found for correlations between rs-FC and melodic structure, this time between left and right HG ROIs. We observed that Fisher's *z* values were positively correlated with regularization behaviors. Individuals with stronger connectivity tend to regularize the signals' melodic contour more (i.e. they tend to reduce contour entropy) (r = 0.43; p=0.001) and to introduce smaller intervals in signals compared to players with weaker rs-FC (this result did not survive Bonferroni correction; p=0.01) (see also *Figure 5—figure supplement 1* for a correlation across all behavioral measures).

## Whole-brain regression analyses on resting state FC

Prediction of behavioral performance by interhemispheric FC extended from auditory to dorsolateral prefrontal regions. *Figure 6* shows that FC between left HG ROI and a cluster including right middle and superior frontal gyri was negatively correlated with melodic regularization (r = −0.58; p<0.001). Individuals who reproduced smoother melodic material in Game 2 tended to have a stronger negative correlation between spontaneous activity in the left primary auditory cortex and right prefrontal regions.

## Digit span

As shown in *Table 1*, digit span scores were not predictive of learning behavior and structural regularization in signaling games (*Table 1*).

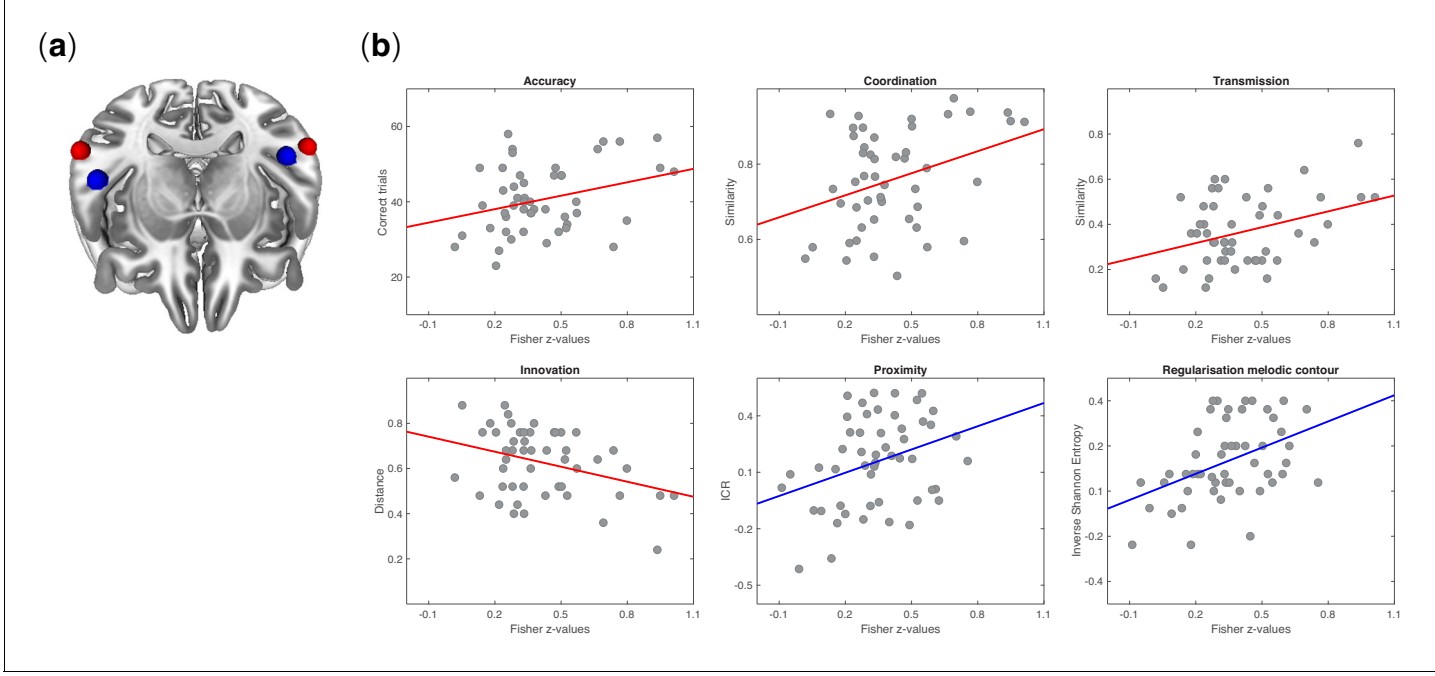

**Figure 5.** Neurobehavioral correlations. (**a**) Tilted axial view of regions activated by the auditory functional localizer task in the posterior STG (red spheres) and anterior STG (blue spheres). (**b**) Pearson's product-moment correlations between rs-FC (among regions activated in the functional localizer) and measures of behavior in signaling games: coordination (r = 0.41; p=0.002), accuracy (r = 0.36; p=0.007), transmission (r = 0.41; p=0.002), innovation (r = −0.41; p=0.003), proximity (r = 0.35; p=0.01), and regularization of melodic contour (r = 0.43; p=0.001). All p-values were Bonferroni corrected at α = 0.05/7 = 0.007. Fisher's z-transformed correlation coefficients were obtained between pairs of 5 mm ROIs from session one and measures of behavior (accuracy, coordination, transmission, innovation) or structural features of tone sequences (proximity, regularization of melodic contours) from the signaling games on session 2. Behavioral measures were defined as the extent to which the code learned by a participant in Game 1 (accuracy and coordination) was faithfully recalled and transmitted in Game 2 (transmission) and reorganized between Games 1–2 (innovation). Each point on a scatterplot is one participant (N = 51).

DOI: https://doi.org/10.7554/eLife.48710.008

The following figure supplement is available for figure 5:

**Figure supplement 1.** Cross-correlation matrix showing.

DOI: https://doi.org/10.7554/eLife.48710.009

## Localizer task-based activity

Percent signal change values obtained for each ROI in the same contrast (DC >SS) did not predict behavior or structural changes in signaling games (all p-values>0.05 uncorrected).

## Task-dependent functional connectivity (PPI)

We measured changes in the functional coupling between seed and target regions during deviant events (C-deviants). Changes in task-dependent functional connectivity did not predict behavior or structural changes in signaling games (no suprathreshold voxel were found; all p-values>0.05 uncorrected).

## Discussion

Using a large sample of participants (N = 51), we showed that rs-FC between homologous portions of the left and right STG predicts individual behavior in learning, transmission, and reorganization of artificial tone systems. These results connect for the first time individual brain characteristics with basic aspects of cultural transmission behaviors in humans (, p. 239). Interestingly, we found that the structural reorganization of melodic patterns was only predicted by rs-FC between the left and right Heschl's gyri, and between left Heschl's gyrus and prefrontal regions, while learning and transmission were only predicted by rs-FC between areas of bilateral planum temporale. This suggests a

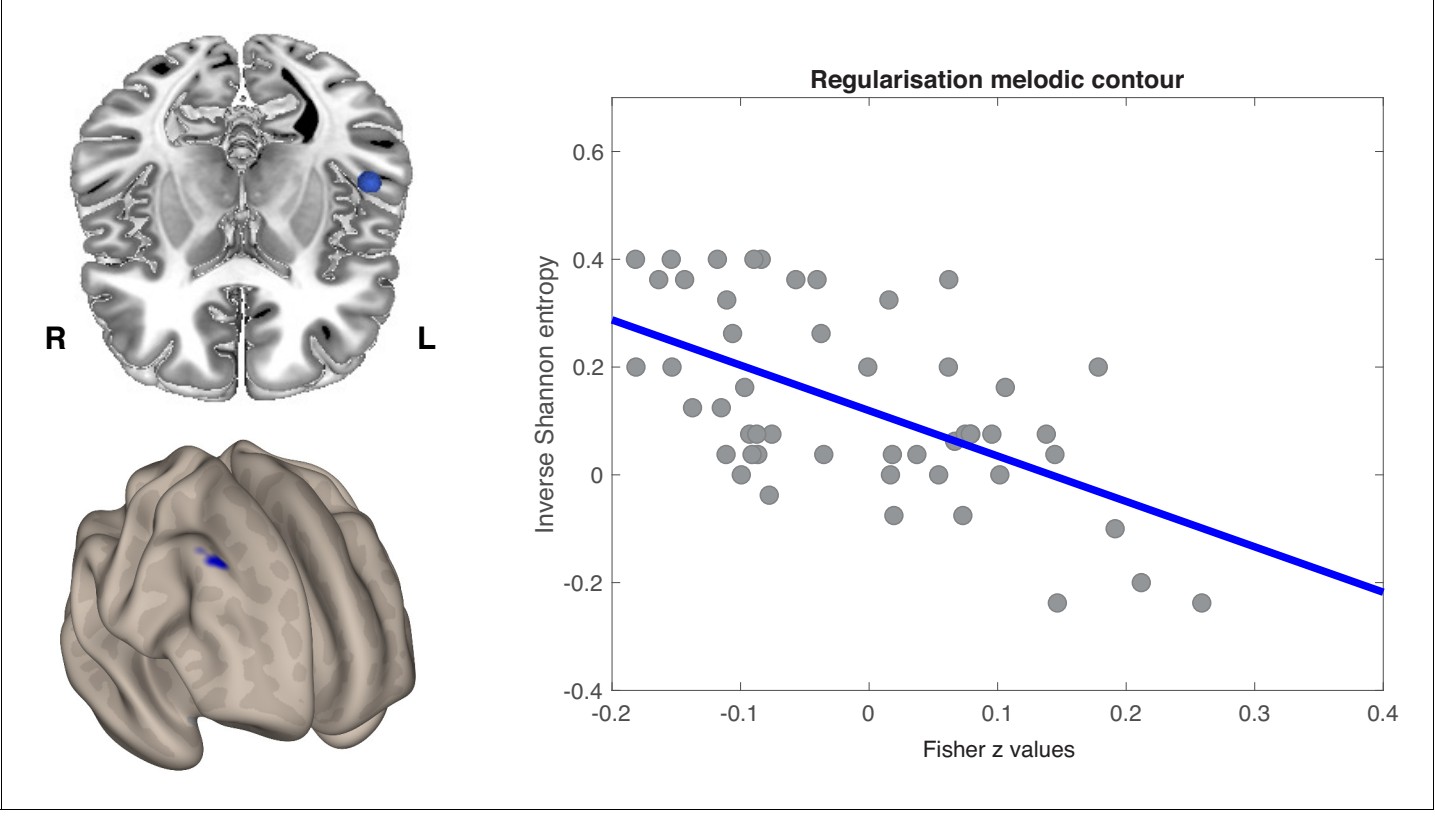

**Figure 6.** Seed-to-voxel regression analysis was conducted over the whole brain. (a) Axial view of the left anterior STG (blue sphere) (top) and the voxel-wise FC correlation map (Fisher Z-transformed) (bottom). One cluster was identified by non-parametric permutation tests as showing a significant negative correlation between temporo-frontal connectivity values and melodic regularization (p<0.05, cluster-mass FWE corrected). (b) Pearson's product moment correlation (r) between melodic regularization and mean FC connectivity between seed and all significant voxels in the frontal cluster (r = −0.58; p<0.001). Each point on a scatterplot is one participant (N = 51).
DOI: https://doi.org/10.7554/eLife.48710.010

*functional dissociation* in the bilateral fronto-temporal auditory network for encoding and regularization of musical material. Task-driven activations did not predict behavior in signaling games.

The bilateral activation in the STG, as evoked by deviant *contour* changes in an oddball functional localizer (*Lumaca and Baggio, 2016*), includes the main cortical generators of the auditory MMN (*Opitz et al., 1999*; *Opitz et al., 2002*; *Wible et al., 2001*; *Schönwiesner et al., 2007*). The MMN reflects auditory change-detection and can therefore be used to study bottom-up processes of auditory scene analysis (ASA), such as the extraction of regularities from acoustic inputs, and melodic and rhythmic grouping during encoding of melodic Gestalts (*Bregman, 1990*; *Koelsch and Siebel, 2005*). The absence of significant activations for deviant *interval* changes is consistent with previous findings suggesting that automatic detection (smaller and slower MMN) in non-musicians is more difficult for pitch interval changes relative to contour changes (*Schiavetto et al., 1999*; *Trainor et al., 2002*), here further hindered by the use of the unfamiliar Bohlen-Pierce scale (*Leung and Dean, 2018*). Unlike some of the previous studies (*Molholm et al., 2005*), we did not observe frontal or parietal activity in pitch deviance detection, which are thought to underlie the generation of the P3a evoked potential (*Soltani and Knight, 2000*). The P3a component typically follows the MMN in time and is related to involuntary attention switching (*Escera et al., 1998*). Our results support the view that automatic detection of deviant sounds and, more generally, pre-attentive operations of ASA are primarily carried out in early auditory cortices (*Schindler et al., 2013*).

The last two decades have seen a surge of interest in the impact of culture on low-level (attention-independent) perceptual processes (*Han and Northoff, 2008*). However, the flip side of that has been largely neglected: namely, whether the functional organization of lower-level sensory systems affects the diversity of cultural traits within (and across) individuals and populations

(*Sperber and Hirschfeld, 2004*). The 'neuronal recycling' hypothesis (*Dehaene and Cohen, 2007*) is a comprehensive view that may also account for the neural origins of cultural variation. Acquired cultural systems and behaviors are accommodated within pre-existing brain networks, initially evolved for different specific adaptive reasons. Such systems and behaviors may then be shaped by the computational constraints inherent to the host networks. Auditory scene analysis (ASA) emerges early in human development (*Winkler et al., 2003*), and it is automatic and widely conserved across species (*Fay, 2008*). ASA neural circuitry might therefore be evolutionarily older than human musicality and may provide some of the neuronal substrates for the cultural recycling of music (*Trainor, 2015*). One prediction of the 'neuronal recycling' hypothesis in the context of auditory perception is that even small differences between individuals in the prior organization of bottom-up ASA auditory circuits, (1) may emerge rapidly in the course of transmission as variants of musical traits and behaviors, and (2) may be amplified in frequency during cultural transmission (*Lumaca et al., 2018b*).

In this work, we support prediction (1) by showing that the inter-hemispheric cortical organization of a temporal auditory network, as revealed by rs-FC, maps onto individual differences in learning and transmission, and in the extent of melodic reorganization in signaling games. A positive correlation was observed between interhemispheric rs-FC in bilateral PT, and the individual accuracy in learning (high coordination, high accuracy) and transmitting (high transmission and low innovation) melodic material in signaling games. *Andoh and Zatorre (2013)* showed that an increase of interhemispheric FC in STG, before and after transcranial magnetic stimulation (TMS), is associated with facilitated encoding and discrimination of melodic material in experimental participants. Increased interhemispheric connectivity in bilateral PT is often associated with superior auditory skills, such as the unique ability to perceive and retain the pitch class of any given tone, or absolute pitch (AP). For example, a graph-theoretical analysis of AP functional data by *Loui et al. (2012)* demonstrated an increased interhemispheric task-related FC between bilateral PT compared to matched controls. This finding was replicated on resting-state data. Although AP is often considered a binary trait, it may be expressed along a continuum in humans (*Bermudez and Zatorre, 2009*). Also, an increased interhemispheric FC in bilateral PT, accompanied by a greater integrity of transcallosal projections, has been found in musicians compared to nonmusicians and is related to perceptual advantages in language-related skills such as phonetic categorization and speech decoding (*Kühnis et al., 2013*; *Elmer et al., 2016*; *Elmer et al., 2017*). These findings support a model of human PT as a computational 'hub' for spectro-temporal analysis of complex auditory objects (*Griffiths and Warren, 2002*). Also, these results favor the view that spontaneous interhemispheric correlation patterns may reflect prior constraints or biases on auditory learning (*Fiser et al., 2010*; *Harmelech and Malach, 2013*).

Two main mechanisms can account for our results. In one model, the coherence of slow spontaneous fluctuations underlying rs-FC may reflect the synaptic efficacy structure of a cortical network, which in turn might modulate the efficiency of information exchange between its functional units during learning (*Varela et al., 2001*; *Harmelech and Malach, 2013*). An alternative, non-exclusive account focuses on the interindividual variability in structural connectivity between bilateral homologous auditory regions, partly mediated by the corpus callosum (CC) (*Westerhausen et al., 2009*). This structural connectivity is strongly associated with FC in healthy individuals (*Honey et al., 2009*; *Lowe et al., 1998*; *Roland et al., 2017*). More specifically, interhemispheric structural connections between homologous regions are mostly subtended by the same slow spontaneous fluctuations in BOLD signals that characterize rs-FC. Regardless of its origins, it has been shown that coordination among brain regions generally decays with the increase of their Euclidean distance, except for bilateral homologous brain regions, where neuronal synchronization remains strong. Resting-state functional connectivity between homologous areas of the two hemispheres has been proposed as a functional organization principle of the human brain (*Salvador et al., 2005b*). More efficient interhemispheric communication between homologous computational units, via either direct (*Johnston et al., 2008*) or indirect structural connections (*Salvador et al., 2005a*), may partly account for the enhanced performance in learning and transmission observed here.

We did not find evidence for a nexus between behavioral performance and task-related brain activations. Intrinsic connectivity properties of the auditory network, more than the extent or the activation level of its functional units, might drive social learning and structural modification of melodic systems. This view is in line with a previous ERP study (*Lumaca and Baggio, 2016*): we showed that only MMN *latencies* were predictive of individual learning, transmission, and

regularization of musical codes in signaling games, and no significant relations were found between behavior and MMN amplitudes. This result was recently extended to the rhythm domain (*Lumaca et al., 2018a*). One proposal is that ERP latencies reflect axonal myelination (*Cardenas et al., 2005*; *Haier et al., 2005*) and diffusion properties (*Tuch et al., 2005*; *Whitford et al., 2011*) of fiber tracts connecting the relevant neural ERP generators, and determining in turn their FC. ERP amplitudes may instead be related to the size and extent of activation of a neuronal population (*Rasser et al., 2011*). This would support the view that spontaneous neuronal synchronization is a more efficient metric of neural information processing than mere task-related activity (*Raichle and Mintun, 2006*).

A limitation of this hypothesis is the possible relation between cultural behaviors and task-evoked FC within the same auditory network described so far. The PPI analysis we performed did not reveal any significant effects. Specifically, oddball-evoked changes in the functional coupling between nodes of the temporal auditory network did not predict individual performance or structural code regularization in signaling games. Up to now, it remains unclear how task and rest FC may relate to each other (*Cole et al., 2014*). One hypothesis is that FC at rest characterizes an 'intrinsic' network architecture that, much like structural connectivity, is stable across multiple brain states (*Fox and Raichle, 2007*; *Vincent et al., 2007*). In contrast, oddball-evoked FC would reveal only a very specific functional state that was unrelated to the behavioral task in our study. Following this view, general constraints imposed on the relevant functional network architecture would mainly be revealed at rest. Our findings contribute to clarifying the nature of the neural mechanisms constraining or driving the cultural evolution of human symbolic systems.

During cultural transmission, information is learned and reproduced across individuals and may become increasingly compressible (low-entropic) and structured and easier to learn and reproduce (*Chater and Vitányi, 2003*). One question now emerging in the music evolution field is whether aspects of compressibility and, in turn, the system-level emergence of musical structure, can be traced back to properties of core auditory mechanisms (*Merker et al., 2015*). Early auditory cortical regions such as HG and PT are critical for encoding and short-term retention of melodic information (*Zatorre et al., 1994*; *Jäncke et al., 2001*; *Leaver et al., 2009*). However, to our knowledge, so far no attempt has been made to investigate dissociations between encoding and compressibility mechanisms, recruited during learning and regularization of cultural material, respectively.

In this study, we provide the first evidence in support of this dissociation. We show that the degree of intrinsic FC within a widely distributed auditory network that includes right dorsolateral prefrontal cortex (DLPFC), in addition to bilateral auditory cortices, is negatively correlated with regularization of melodic material during memory retrieval. Individuals with a stronger negative correlation (antiphase coherence) between DLPFC and auditory cortices introduce smoother forms in the melodic set. This result was not predicted, and the nature of the underlying neuronal processes remains to be described in detail. However, this is consistent with theoretical accounts of symbolic regularization (*Kam and Newport, 2009*) and with neuropsychological data (e.g. congenital amusia). One hypothesis relates to the crucial role of dorsolateral prefrontal structures in working memory (*Rypma and D'Esposito, 1999*). A process-specific model has been proposed in past studies, where DLPFC supports maintenance, monitoring, and manipulation of perceptual representations stored in more posterior sensory regions (see for a support in the auditory domain). A reduced connectivity between DLPFC and sensory regions may affect the quality of mnemonic information available for later retrieval, resulting in a loss of details and on a greater emphasis on global features. The connectivity pattern we observed in 'innovators' (i.e. the individuals regularizing the melodic material more), echoes previous findings in amusics. A reduced frontotemporal connectivity was observed in this population, either at rest (*Leveque et al., 2016*) or during the active and passive listening of auditory material (*Albouy et al., 2013*; *Hyde et al., 2011*), and is linked to impaired melodic retrieval (*Albouy et al., 2015*). The auditory cortices were also found to be overconnected compared to control groups (*Leveque et al., 2016*). This converging evidence suggests that an increased projection of acoustic information between bilateral auditory regions, and decreased frontotemporal projection, can reveal poorer auditory processing capabilities in individuals. Functional decoupling between prefrontal structures and sensory regions, and an over coupling of homologue auditory areas, can be thus invoked as one putative mechanism to explain regularization in the acquisition of cultural material (*Thompson-Schill et al., 2009*; *Kam and Newport, 2009*). We showed that even small differences in the functional architecture at rest of temporo-frontal networks can be

manifested in regularization behaviors, already in a single round of learning and transmission. These differences can be amplified by cultural transmission and can produce population-level effects, as shown by computational (*Kirby, 2001*; *Griffiths et al., 2008*), behavioral research (*Lumaca and Baggio, 2017*; *Ravignani et al., 2016*), and population-genetics studies (*Dediu and Ladd, 2007*).

Our findings point to degrees of interhemispheric functional connectivity in the relevant sensory networks as a possible neural source of inter-individual variation in three main aspects of human cultural behavior: (i) social learning, (ii) symbolic transmission, and (iii) structural modification of symbolic systems. To our knowledge, this may be the first demonstration of the existence of specific neurophysiological mechanisms underlying perceptuo-cognitive biases or constraints in music evolution. Inter-individual variability in these biases or constraints may give rise to system-level variation in the organization, transmission, and acquisition of human musical systems.

## Acknowledgements

The authors thank Martin Dietz for useful comments, Hella Kastbjerg for proofreading the manuscript, and Claudia Iorio, Ulrika Varankaite, Eira Aksnes, and Malou Ryborg for assistance in data acquisition. Center for Music in the Brain is funded by the Danish National Research Foundation (DNRF117).

## Additional information

### Funding

| Funder | Grant reference number | Author |
| --- | --- | --- |
| Danmarks Grundforsknings-fond | DNRF117 | Peter Vuust |

The funders had no role in study design, data collection and interpretation, or the decision to submit the work for publication.

### Author contributions

Massimo Lumaca, Conceptualization, Data curation, Formal analysis, Investigation, Methodology, Writing—original draft, Writing—review and editing; Boris Kleber, Formal analysis, Visualization, Methodology; Elvira Brattico, Conceptualization, Funding acquisition, Writing—review and editing; Peter Vuust, Conceptualization, Supervision, Funding acquisition, Writing—review and editing; Giosue Baggio, Conceptualization, Supervision, Visualization, Writing—review and editing

### Author ORCIDs

Massimo Lumaca https://orcid.org/0000-0002-3432-3911
Giosue Baggio https://orcid.org/0000-0001-5086-0365

### Ethics

Human subjects: The study was approved by the ethics committee of the Central Denmark Region (nr. 1083). All participants gave their written consent and filled out an fMRI safety form before the experiment.

### Decision letter and Author response

Decision letter https://doi.org/10.7554/eLife.48710.015
Author response https://doi.org/10.7554/eLife.48710.016

## Additional files

### Supplementary files

• Transparent reporting form DOI: https://doi.org/10.7554/eLife.48710.011

## Data availability

Data files and scripts for fMRI, resting-state functional connectivity, and signaling games have been deposited in Dryad Digital Repository (https://doi.org/10.5061/dryad.2jj01c1).

The following dataset was generated:

| Author(s) | Year | Dataset title | Dataset URL | Database and Identifier |
|---|---|---|---|---|
| Lumaca M, Kleber B, Brattico E, Vuust P, Baggio G | 2019 | Datafiles and scripts from: Functional connectivity in human auditory networks and the origins of variation in the transmission of musical systems | https://doi.org/10.5061/dryad.2jj01c1 | Dryad Digital Repository, 10.5061/dryad.2jj01c1 |

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
