## [Decision Letter]

Thank you for submitting your article "Functional connectivity in human auditory networks and the origins of variation in the transmission of musical systems" for consideration by *eLife*. Your article has been reviewed by three peer reviewers, and the evaluation has been overseen by a Reviewing Editor and Barbara Shinn-Cunningham as the Senior Editor. The following individual involved in review of your submission has agreed to reveal their identity: Lutz Jäncke (Reviewer #2).

The reviewers have discussed the reviews with one another and the Reviewing Editor has drafted this decision to help you prepare a revised submission.

This novel study addresses whether individual approaches to a lab model of cultural learning and transmission were reflected in resting state connectivity, and demonstrates a number of interesting and specific correlations.

The reviewers enjoyed reading this work and have raised a number of points related to exposition and interpretation that would increase its impact and accessibility. We invite the authors to submit a resubmission that addresses the points.

Essential revisions:

1) The rationale for choosing the studied signaling game as a model of transmission of a musical system should be better explained. It appears overstated that it reflects "learning, transmission, and the structural modification of an artificial musical system" (as stated in the Abstract). More precisely, what does "musical system" mean in the present study design? Participants have to learn (and then transmit) a code linking emotions (on faces) and short melodies (constructed from the Bohlen-Pierce scale). There does not appear to be underlying rules to be learned regarding melody construction or to infer regarding the pairwise associations between emotions and melodies. It therefore appears as mere associative audio-visual learning, with more or less accurate reproduction of the learned melodies depending on the participants.

2) A concern was possible confounds like general intelligence or g. No participant neuropsychological data are provided and an issue is whether correlations might exist between more general measures of perception and neuropsychological function. We would be interested to hear the author's comments. The authors might also include more behavioural data on musicianship.

3) The results were quite difficult to follow because the Materials and methods follow. Either put the Materials and methods section first or help the reader along more in Results section.

4) More detail on methodology for correlation coefficients is requested. It was not very clear from the text exactly which correlations between behavioural measures and functional connectivity measures were tested (how many tests were performed?). One also wonders how much the different behavioural measures correlate between them. Furthermore, there is no report of the non-significant correlations, so overall it is difficult to appreciate the full data pattern. Also, I did not understand why only percent signal changes are analysed from the localizer data, connectivity measures obtained during listening would also be relevant here.

5) Was any analysis of functional connectivity between the left and right auditory cortex produced by activation carried out? The main interest here is in resting state connectivity likely reflecting structural connections, and the functional localisers use a very different task to the transmission game but functional connectivity between auditory cortex during activation might also be related to game performance.

6) It was interesting that structural reorganisation of melody was predicted by resting connectivity between right and left HG which might be expected at a higher level in auditory hierarchy.

7) The negative correlation between regularisation behaviour and auditory and frontal resting connectivity is unlikely to have been predicted and the discussion of this is cagey. Further comments would be welcome.

8) The relationship between resting state connectivity in right and left PT and learning/transmission made sense in term of literature on involvement of PT in auditory learning which might be discussed. In general, there are several recent and older papers demonstrating that functional and structural connectivity (also in the auditory system) is related to superior cognitive performance (e.g., IQ: Langer et al., 2012), synesthesia (e.g., Jäncke and Langer, 2012) and also to musicianship (e.g., absolute pitch: Jäncke et al., 2012; Loui et al., 2012; Loui et al., 2011; Hamm et al., 2013; but also musicianship in general: Klein et al., 2016, Elmer et al., 2016). Thus the idea that functional connectivity is a marker of superior performance is not entirely new although the authors focus more on a particular musical aspect. In addition, functional and structural connectivity features are often associated with „negative" behavioral aspects (e.g., depression, anxiety, and cardiac problems; there are many papers of this type).

---

## [Author Response]

Essential revisions:1) The rationale for choosing the studied signaling game as a model of transmission of a musical system should be better explained. It appears overstated that it reflects "learning, transmission, and the structural modification of an artificial musical system" (as stated in the Abstract).

We agree with the reviewers that this point remained unclear in our previous formulation, and we thank them for raising this issue. The signaling games used in our study is a tool to (formally) model and (empirically) investigate three specific aspects of cultural transmission: social learning, transmission, and structural modification of learned codes. Our hypothesis is that there may exist a link between individual differences in these three cultural behaviors and inter-individual neural variability. Second, we did not actually (claim we) use full-fledged realistic musical systems (see below).

We have scanned the text for any potential ambiguities in this respect, and we have edited all passages that contained correspondingly unclear statements.

More precisely, what does "musical system" mean in the present study design?

We acknowledge that our use of ‘musical systems’ may be incorrect in some parts of the manuscript. This term typically refers to the complex set of rules and forms that define the musical material of a given culture. In order to improve clarity, we have replaced ‘artificial musical systems’ with ‘artificial tone systems’. We define ‘artificial tone systems’ as an organized system of tones that is novel for all participants in the experiments (Loui, 2007).

The term ‘musical system’ has been maintained in a few places in the manuscript where it is more appropriate (including the title). As artificial languages are generally assumed to be relevant to understand natural languages (Kirby et al., 2008), artificial tone systems can be used to study actual musical systems (Lumaca and Baggio, 2017; Ravignani, Delgado and Kirby, 2016).

Participants have to learn (and then transmit) a code linking emotions (on faces) and short melodies (constructed from the Bohlen-Pierce scale). There does not appear to be underlying rules to be learned regarding melody construction or to infer regarding the pairwise associations between emotions and melodies. It therefore appears as mere associative audio-visual learning, with more or less accurate reproduction of the learned melodies depending on the participants.

In signaling games, there are no rules underlying the construction of the initial material (or “seeding material”). Principles of organization (e.g., proximity and continuity) are expected to emerge by a combination of cognitively-driven transmission errors and, partly, by processes of social interaction, such as coordination (game 2). In this respect, signaling games are more complex than a mere associative audio-visual learning task.

2) A concern was possible confounds like general intelligence or g. No participant neuropsychological data are provided and an issue is whether correlations might exist between more general measures of perception and neuropsychological function. We would be interested to hear the author's comments. The authors might also include more behavioural data on musicianship.

We agree that this is a possibility. In our study, we needed a control test for this type of general confound while maintaining the duration of the fMRI experiment at a reasonable length (1.5h including preparation, recording and debriefing). We opted for the Digit Span, a subtest of the Wechsler Adult Intelligence Scale (WAIS) and the Wechsler Memory Scales (WMS) that measures attention efficiency and capacity (forward span) and WM (backward span).

The results of this test are now included in a new table (Table 2). No significant correlations were observed between digit span scores and behavioral scores (for a similar result, see Lumaca and Baggio, 2016). This suggests that individual differences in auditory cortex function, rather than other aspects of neuropsychological function, constrained the learning and transmission of artificial tone systems.

Regarding experience-related effects on brain connectivity:

Thank you for reminding us to address this point. We need to mention that more than half of our participants reported 0 years of formal musical training with an instrument (n=32). Inter-individual variability in musical training seems too low to drive the observed inter-individual variability in brain connectivity (see below). We also doubt that few months of musical training may significantly ‘affect’ the functional relationship between auditory cortices. The average number of years of musical training across all participants is 0.6 (STD=0.98; n=51). We have now added these statistics in the Materials and methods section.

We computed the Pearson’s correlation between number of years of musical training and connectivity measures. No correlation survived the Bonferroni correction.

Though beyond the scope, the point raised by the reviewers is interesting. However, to be properly addressed, it would require a ‘full battery’ of music-psychological tests (e.g., GOLD-MSI). This was not done in the current study, since we tried to maintain the experiment <1.5h.

3) The Results were quite difficult to follow because the Materials and methods follow. Either put the Materials and methods section first or help the reader along more in Results section.

Following the reviewers’ suggestion, we have now moved the Materials and methods section before the Results section.

4) More detail on methodology for correlation coefficients is requested. It was not very clear from the text exactly which correlations between behavioural measures and functional connectivity measures were tested (how many tests were performed?). One also wonders how much the different behavioural measures correlate between them. Furthermore, there is no report of the non-significant correlations, so overall it is difficult to appreciate the full data pattern. Also, I did not understand why only percent signal changes are analysed from the localizer data, connectivity measures obtained during listening would also be relevant here.

We have now included a new table (Table 2), showing the whole pattern of correlation results. In doing so, we have noticed we had missed one significant correlation in the previous manuscript (i.e., between innovation and lHG-lSTG rs-FC). We included this result in the new version.

By including the digit span test, we have also changed the Bonferroni correction. For each behavioral measurement, we performed 7 independent tests (6 is the number of ROI-to-ROI combinations, plus 1 test with the digit span) (see Table 2).

We included a supplementary figure (Figure 5—figure supplement 1) showing correlations across all behavioral measures.

In regard to the comment on functional connectivity analyses, please read below.

5) Was any analysis of functional connectivity between the left and right auditory cortex produced by activation carried out? The main interest here is in resting state connectivity likely reflecting structural connections, and the functional localisers use a very different task to the transmission game but functional connectivity between auditory cortex during activation might also be related to game performance.

We agree on this point with the reviewers. We had a specific interest in limiting our connectivity analysis to resting state. However, we can use task-based functional connectivity as a control. We have now performed a psychophysiological interaction (PPI) analysis for each seed ROI in combination with a whole-brain second-level regression analysis, using behavioral measures as covariates of interest. We also implemented a seed-to-ROI approach (the analysis reported in the manuscript). No voxel survived the statistical threshold in either analysis. This suggests that constraints imposed on the auditory functional architecture, possibly operating during learning in signaling games, are primarily revealed at rest. We have added this interpretation in the Discussion.

Psychophysiological interaction (PPI) analyses are now included in the Materials and methods and Results sections.

6) It was interesting that structural reorganisation of melody was predicted by resting connectivity between right and left HG which might be expected at a higher level in auditory hierarchy.

The reviewers makes an interesting point, yet previous research has shown no evidence for an exclusive link between structural regularization and higher-level auditory systems. Instead, neuroimaging and neurophysiological studies show that tonal and melodic processing can occur already in HG (Schindler et al., 2013; Zatorre and Zarate, 2010), and that primary auditory cortex can be involved, together with higher level areas, in complex auditory operations, such as melodic retrieval and manipulation (e.g., Albouy et al., 2017). Also, regularization in this study was not confined to HG, but was linked to the rs-FC of a fronto-temporal network that includes dorsolateral prefrontal cortex (please see our reply to the next comment).

7) The negative correlation between regularisation behaviour and auditory and frontal resting connectivity is unlikely to have been predicted and the discussion of this is cagey. Further comments would be welcome.

We agree with the reviewers’ point. The seed-to-voxel analysis was exploratory and, contrary to the ROI-to-ROI analysis, we did not have specific expectations. However, our results seem to support a theoretical account of code regularization (Hudson Kam, Newport, 2009). Also, they are in line with previous neuropsychological data. In our study, regularization was predicted by greater connectivity between auditory cortices (HG) and by diminished connectivity between Heschl’s gyri and prefrontal structures. A similar resting-state connectivity pattern is observed in the amusic brain (Leveque et al., 2016) and may be linked to stronger individual constraints in auditory processing. This would in turn create pressure to regularize the melodic material (Hudson Kam, Newport, 2009). We have now included this point in the Discussion section.

8) The relationship between resting state connectivity in right and left PT and learning/transmission made sense in term of literature on involvement of PT in auditory learning which might be discussed. In general, there are several recent and older papers demonstrating that functional and structural connectivity (also in the auditory system) is related to superior cognitive performance (e.g., IQ: Langer et al., 2012), synesthesia (e.g., Jäncke and Langer, 2012) and also to musicianship (e.g., absolute pitch: Jäncke et al., 2012; Loui et al., 2012; Loui et al., 2011; Hamm et al., 2013; but also musicianship in general: Klein et al., 2016, Elmer et al., 2016). Thus the idea that functional connectivity is a marker of superior performance is not entirely new although the authors focus more on a particular musical aspect. In addition, functional and structural connectivity features are often associated with „negative" behavioral aspects (e.g., depression, anxiety, and cardiac problems; there are many papers of this type).

We thank the reviewers for this helpful suggestion. We now succinctly discuss the relationship between interhemispheric connectivity in bilateral PT and auditory processing (see Discussion section).